# Cell Death Signaling Pathway Induced by Cholix Toxin, a Cytotoxin and eEF2 ADP-Ribosyltransferase Produced by *Vibrio cholerae*

**DOI:** 10.3390/toxins13010012

**Published:** 2020-12-24

**Authors:** Kohei Ogura, Kinnosuke Yahiro, Joel Moss

**Affiliations:** 1Advanced Health Care Science Research Unit, Institute for Frontier Science Initiative, Kanazawa University, Kanazawa 920-0942, Japan; 2Department of Molecular Infectiology, Graduate School of Medicine, Chiba University, Chiba 260-8670, Japan; 3Pulmonary Branch, National Heart, Lung, and Blood Institute, National Institutes of Health, Bethesda, MD 20892-1590, USA; mossj@nhlbi.nih.gov

**Keywords:** bacterial cytotoxin, ADP-ribosyltransferase, mono-ADP-ribosylation, cell death, apoptosis, hepatocytes

## Abstract

Pathogenic microorganisms produce various virulence factors, e.g., enzymes, cytotoxins, effectors, which trigger development of pathologies in infectious diseases. Cholera toxin (CT) produced by O1 and O139 serotypes of *Vibrio cholerae* (*V. cholerae*) is a major cytotoxin causing severe diarrhea. Cholix cytotoxin (Cholix) was identified as a novel eukaryotic elongation factor 2 (eEF2) adenosine-diphosphate (ADP)-ribosyltransferase produced mainly in non-O1/non-O139 *V. cholerae*. The function and role of Cholix in infectious disease caused by *V. cholerae* remain unknown. The crystal structure of Cholix is similar to *Pseudomonas* exotoxin A (PEA) which is composed of an N-terminal receptor-recognition domain and a C-terminal ADP-ribosyltransferase domain. The endocytosed Cholix catalyzes ADP-ribosylation of eEF2 in host cells and inhibits protein synthesis, resulting in cell death. In a mouse model, Cholix caused lethality with severe liver damage. In this review, we describe the mechanism underlying Cholix-induced cytotoxicity. Cholix-induced apoptosis was regulated by mitogen-activated protein kinase (MAPK) and protein kinase C (PKC) signaling pathways, which dramatically enhanced tumor necrosis factor-α (TNF-α) production in human liver, as well as the amount of epithelial-like HepG2 cancer cells. In contrast, Cholix induced apoptosis in hepatocytes through a mitochondrial-dependent pathway, which was not stimulated by TNF-α. These findings suggest that sensitivity to Cholix depends on the target cell. A substantial amount of information on PEA is provided in order to compare/contrast this well-characterized mono-ADP-ribosyltransferase (mART) with Cholix.

## 1. Introduction

*Vibrio cholerae* (*V. cholerae*), a gram-negative bacterium with a curved rod shape, belongs to the Vibrionaceae family and colonizes shellfish [1,2,3] and fish [4,5] in aquatic environments such as coastal salt waters and estuaries. *V. cholerae* is categorized into more than 206 serogroups [6]. Serotypes O1 and O139 of *V. cholerae* strains possess two main virulence gene sets: those of cholera toxin (CT) and the toxin-coregulated pilus, which cause an acute watery diarrheal disease. Recent reports showed that non-O1/O139 *V. cholerae* (NOVC) strains also cause diarrhea [7]. The gene for CT is encoded by a filamentous bacteriophage [8]. Some NOVC strains produce CT, which is implicated in cholera-like illnesses [9,10,11]. Vezzuli et al. noted an increasing number of case reports showing bacteremia and fatalities involving sepsis and necrotizing fasciitis apart from the diarrhea associated with NOVC strains [7,12]. Patients with liver cirrhosis were found to be susceptible to NOVC bacteremia [13,14].

Other than CT, some *V. cholerae* strains possess various virulence factors, e.g., heat-stable toxin [15], hemolysin [16], type III secretion system [17,18], multifunctional autoprocessing repeats-in-toxin [19], accessory toxins like zonula occludens toxin [20,21], Cholix toxin [22,23]. Cholix toxin (Cholix) belongs to a mono-ADP-ribosyltransferase (mART) family, as does CT. While CT ADP-ribosylates the alpha subunit of the heterotrimeric G protein Gs, Cholix specifically targets diphthamide of eukaryotic elongation factor 2 (eEF2), a mechanism similar to that used by diphtheria toxin (DT) and *Pseudomonas* exotoxin A (PEA) from, respectively, *Corynebacterium diphtheriae* and *Pseudomonas aeruginosa* [22]. In addition to the three mARTs (i.e., DT, PEA, and Cholix), Fernández-Bravo et al. reported recently that some *Aeromonas hydrophila* strains possess exotoxin A genes [24]. The four mARTs catalyze ADP-ribosylation of the diphthamide residues using nicotinamide adenine dinucleotide (NAD^+^) as the ADP-ribose donor [25]. Eukaryotic elongation factor 2 catalyzes the translocation of mRNA and peptidyl-tRNA; inactivation of eEF2 by ADP-ribosylation results in inhibition of protein synthesis, followed by host cell death [26,27].

## 2. *V. cholerae* Strains Possessing *Cholix* Genes

The *cholix* genes were found in *V. cholerae* strains isolated in various countries, such as Bangladesh [4,23,28], Germany [29,30,31,32], India [28], Iran [33], Kenya [34], Mexico [23,29,35], and the United States [23]. While the *cholix* gene was present in O1 serotype strains [4,23], some researchers proposed that the gene appears to be preferentially associated with NOVC strains, with higher prevalence in NOVC than O1/O139 *V. cholerae* strains [28,29]. It remains unknown how the *cholix* gene was transferred to some *V. cholerae* chromosomes. The guanine-cytosine (GC) content of the *cholix* gene from a *V. cholerae* TP strain [36] (44%) is significantly lower than that of toxA (66%), which encodes *Pseudomonas* exotoxin A (PEA), thus strongly suggesting that the *cholix* gene is not the result of a recent lateral transfer from *P. aeruginosa* [23]. No other indicators of lateral gene transfer, such as phage-like sequences or insertion sequence elements, are seen near this region of the *V. cholerae* genome [23]. The IslandViewer4 web tool showed that the *cholix* gene is not designated to any genomic island [37]. The two genes surrounding the *cholix* gene were highly conserved in the chromosomes of *cholix*-negative *V. cholerae* strains. These results suggested that the probability of horizontal transfer is low.

## 3. Structural Insights

The crystal structure of Cholix was first reported by the Merrill group [22]. Their studies summarized structural insights into Cholix [38]. Based on the Cholix/NAD^+^ complex structure and site-directed mutagenesis, Fieldhouse et al. proposed a new kinetic model for NAD^+^ binding and mART activity by Cholix [39]. Next, Turgeon et al. tested various chemical compounds which can inhibit cytotoxicity of Cholix by competitive binding to the NAD^+^ binding site in Cholix [40]. In order to propose structure/function relationships of the Cholix catalytic domain, the group conducted in silico simulation of molecular dynamics [41,42]. Their findings revealed structural and enzymatic characteristics of Cholix as an eEF2-mART.

The crystal structure of a new exotoxin from *Aeromonas hydrophila* resembles those of PEA and Cholix [43] (Figure 1). The primary amino acid sequence of *Aeromonas* exotoxin A exhibited, respectively, 64% and 35% identity with PEA and Cholix, while its C-terminal mART domain had, respectively, 72% and 46% identity with amino acid sequences of PEA and Cholix [43]. As shown in Figure 1, charged amino acid residues on the surfaces of the three mART toxins are different. The structures of the NAD^+^-binding pockets in the C-terminal domains also vary. While Cholix possesses an EDETV sequence around the catalytic Glu581 (underlined), PEA and *Aeromonas* exotoxin A have an RLETI sequence [43]. Jørgensen et al. reported that the *K*_m(NAD_^+^_)_ of Cholix is lower than that of PEA [22].

## 4. Cholix Receptor

The low-density lipoprotein receptor-related protein 1 (LRP1) is a specific PEA receptor [45]. Jørgensen et al. reported that sensitivity to Cholix is significantly lower in embryonic fibroblasts derived from an LRP1-deficient (LRP−/−) mouse [22]. Consistent with that report, we also found in HeLa cells that Cholix-induced cell death was partially suppressed by LRP1 knockdown (Figure 2). Therefore, LRP1 is considered to be a receptor. However, Jørgensen et al. also reported that the LRP1-deficient cells still showed some sensitivity to Cholix [22]. The suppression of Cholix-induced cell death by LRP1 knockdown was only partial. We also reported that Cholix-induced apoptosis was not suppressed at all by LRP1 knockdown in human hepatocytes [46]. Further, Cholix induced significant cell death in human colon cell line HCT116 cells, whose *LRP1* gene was mutated by frameshifts [47,48]. These reports indicate that there may be another pathway for intoxication independent of LRP1.

## 5. Translocation of Cholix in Host Cells

In host cells, Cholix is transported from endosomes to the endoplasmic reticulum and then reacts with eEF2 in the cytosol. PEA follows a similar intracellular path. In the acidic, early-endosomal environment, PEA is cleaved into two fragments by the protease furin, which digests a furin motif (RHRQPRG) [49,50] to generate a 28 kDa N-terminal domain and a 37 kDa C-terminal mART domain. Protein disulfide-isomerase further reduces the disulfide bond in PEA, although the disulfide bond is likely reduced in the endoplasmic reticulum (ER) [51]. Similar to PEA, Cholix has the furin cleavage site (RHKR) and the disulfide bond [22]. Cholix reduction in cell viability was suppressed in the presence of a furin inhibitor, decanoyl-Arg-Val-Lys-Arg-chloromethylketone (Figure 3). These findings suggest that cleavage of the furin site is required for activation of Cholix as well as of PEA. However, Morlon-Guyot et al. found that full-length PEA was also translocated into the cytosol in a mouse fibroblast cell line (L929 cells) [50]. It remains unclear whether Cholix, similarly to PEA, is translocated into the cytosol without processing in L929 cells.

After transportation through late endosomes into the trans-Golgi network [52], the C-terminal mART domain of PEA is translocated to the ER, mainly via the KDEL receptor-mediated pathway [53]. The C-terminal motif REDLK of the mART domain was cleaved at the lysine residue (REDLK) by a carboxypeptidase of the host cell surface [54]. Then, the exposed REDL motif is able to bind to a KDEL receptor, which participates in an important step of the intracellular trafficking pathway, followed by translocation to the ER in a retrograde process [55,56]. Finally, the mART domain of PEA utilizes the cellular ER-associated protein degradation (ERAD) pathway to move from the ER into the cytosol [57,58,59]. Toxins that follow the ERAD pathway to the cytosol have an arginine over lysine amino acid bias to avoid ubiquitination and degradation by the proteasome [60]. While the bias was also observed in the mART domain of Cholix (26 arginines and 12 lysines), it was lower than that of PEA (30 arginines and 3 lysines). Cholix also possesses the C-terminal KEDL motif (RKDELK). These data suggested that translocation of Cholix follows a similar pathway to that of PEA. Awasthi et al. found that a Cholix variant with HDELK in place of RKDELK did not show cytotoxicity in mice [28]. However, KDEL receptors recognize both KDEL and HDEL sequences [61,62,63,64,65]. It remains unclear why the Cholix HDELK variant lacked cytotoxicity.

In addition, by immunoblotting using an anti-Cholix antibody, we found that Cholix was detected in the mitochondrial fraction rather than in the cytosolic compartment, suggesting that Cholix can be translocated into mitochondria; however, there is no information on the translocation mechanism [46].

## 6. ADP-Ribosylation of eEF2 by Cholix

Activated Cholix specifically ADP-ribosylates the diphthamide synthesized from a histidine residue (His715 in mammalian cells) in eukaryotic elongation factor 2 (eEF2) by the diphthamide biosynthesis pathway [66,67] (Figure 4). During protein synthesis, eEF2 translocates mRNA from the ribosomal A-site to the P-site and mediates the elongation step of protein synthesis [66,68]. Cholix-induced eEF2-ADP-ribosylation inhibits ribosomal protein synthesis, resulting in the death of host cells.

## 7. Liver Damage Induced by Cholix

Previous studies demonstrated the effects of Cholix in vivo. Cholix did not cause fluid accumulation in the rabbit ileal loop assay [28]. Intraperitoneal administration to mice of Cholix, but not the catalytically inactivated mutant Cholix (E581A), resulted in death with severe hepatotoxicity [28,70]. Histological analysis of Cholix-injected mice showed hemorrhagic lesions with necrotic, apoptotic, and inflammatory cells, especially in the liver zone between the Glisson’s sheath and the central vein, by periodic acid-Schiff (PAS) and hematoxylin-eosin (HE) staining. Further, liver damage marker alanine transaminase (ALT) was significantly increased in Cholix-injected mice [70], consistent with the conclusion that liver is a target organ of Cholix. PEA also caused liver damage [71], which involved liver macrophage-like Kupffer cells that produced tumor necrosis factor-α (TNF-α) [72,73]. IL-18 and perforin also participated in the liver damage seen in PEA-injected mice [74]. Further, following PEA intoxication of mice, Kupffer cell-dependent, early TNF-α production required T cells [72].

Previously, we analyzed Cholix-induced cell death of human epithelial hepatoblastoma-derived (HepG2) liver cells and immortalized human hepatocytes [70]. HepG2 cells showed slightly decreased cell viability in the presence of Cholix. Addition of TNF-α with Cholix to HepG2 cells significantly enhanced activation of caspases, resulting in poly (ADP-ribose) polymerase (PARP) cleavage followed by increased cytotoxicity. Inhibition of TNF-α/Cholix-activated JNK or ERK by a specific inhibitor (SB20350 or U0126) suppressed PARP cleavage. Cholix-induced PARP cleavage was enhanced in the presence of PKC activator phorbol 12-myristate 13-acetate (PMA) as well as by TNF-α and suppressed by the PKC inhibitor even in the presence of TNF-α. Further, ROS inhibitor N-acetyl cysteine suppressed JNK activation, which partially inhibited the TNF-α/Cholix apoptosis-signaling pathway. These findings suggest that TNF-α is an enhancer of Cholix-induced apoptosis, which was involved in ROS generation, PKC activation, and MAPK activation. On the other hand, immortalized human hepatocytes were more sensitive to Cholix compared to HepG2 cells. In hepatocytes, Cholix-induced apoptosis was mediated through severe mitochondrial damage that was not promoted by addition of TNF-α. Neither the JNK nor the ERK inhibitor (SB20350 or U0126) suppressed Cholix-induced apoptosis.

There are a few reports regarding NOVC-related liver disease [75,76]. However, it is not clear if these NOVC strains have Cholix. In addition, the pathway used by Cholix to translocate from the gastrointestinal tract to the liver remains unknown. Interestingly, Taverner et al. recently reported that catalytically inactive mutant Cholix (E581A) and its N-terminal domain can be transported across human intestinal epithelia in vitro (confluent monolayers of human small intestinal tissues) and rat jejunum in vivo by apical to basal transcytosis [77]. This apical-to-basal transcytosis pathway utilizes a vesicular trafficking mechanism, independent of intoxication by mART.

## 8. Other Cell Death Mechanisms

In addition to hepatocytes, cytotoxicity of Cholix was also tested with other cells, such as mouse fibroblasts [22], intestinal cell lines (i.e., Caco-2, HCT116, and RKO) [47], and HeLa cells [28,47]. Although Cholix did not cause fluid accumulation in rabbit ileal loop assays [28], this toxin exhibited significant cytotoxicity to the intestinal cell lines [47]. Intestinal cell death by Cholix was not affected by the presence of a general caspase inhibitor (Z-VAD-FMK), indicating that intestinal cell death was not a result of apoptosis.

HeLa cells are a commonly used epithelial cell line derived from human cervical cancer. Cholix-treated HeLa cells showed decreased viability, which was significantly repressed by Z-VAD-FMK [47]. In HeLa cells, Cholix stimulated an apoptotic pathway dependent on activation of inflammatory caspases (caspase-1, -4, and -5), followed by mitochondrial outer membrane permeabilization that occurred through rapid degradation of the anti-apoptotic Bcl-2 family protein Mcl-1 and conformational changes of pro-apoptotic Bcl-2 family members, Bak and Bax.

Mcl-1 is a protein with a short half-life, which is rapidly lost in the presence of the protein synthesis inhibitor, cycloheximide [78]. In addition, the loss of Mcl-1 induced by Cholix was suppressed in the presence of proteasome inhibitor MG132. Thus, Cholix decreased Mcl-1 content by inhibition of protein synthesis combined with rapid proteasomal degradation and promoted conformational changes of pro-apoptotic Bcl-2 family members, Bak and Bax [47].

## 9. Prohibitin Binding and Mitochondrial Dysfunction

Recently, we found by immunoprecipitation that Cholix interacts with prohibitin (PHB) 1 and 2 on the cell surface [46]. PHBs are ubiquitously expressed in various cells and mainly localized in mitochondria, but also expressed in nuclei and cell membranes [79]. PHBs contribute to mitochondrial biology [79] and homeostasis [80,81] by regulating ROS formation, protein degradation, mitochondrial morphology, and the oxidative phosphorylation (OXPHOS) complex [82]. The expression of PHB1 was decreased by Cholix in hepatocytes. PHB deficiency impairs respiratory supercomplex formation [83] and causes overproduction of reactive oxygen species (ROS) [84]. In agreement with previous studies, Cholix-induced PARP cleavage, ROS generation, and apoptotic chromatin assembly were significantly increased in PHB knockdown cells compared with control cells. In PHB-overexpressing cells, Cholix-induced apoptotic signaling was inhibited [46]. These findings suggest that PHB is a Cholix interaction protein, not its receptor. Cholix-dependent reduction in PHB1 expression triggers PHB dysfunction, which might lead to altered mitochondrial respiratory supercomplexes, followed by promotion of ROS production.

Rho-associated, coiled coil-containing protein kinase protein 1 (ROCK1) is a crucial factor in mitochondrial fission and ROS generation [85] and is a direct cleavage substrate of activated caspase-3, which is involved in apoptosis [86]. ROCK1 knockdown and ROCK inhibitor Y27632 suppressed Cholix-induced cleavage of PARP and ROCK1 and ROS generation [46]. In PHB knockdown cells, Cholix increased ROCK1 cleavage, followed by enhanced apoptosis, plasma membrane blebbing, and ROS generation, suggesting that ROCK1 participates in a Cholix-induced apoptotic pathway [46].

## 10. Conclusions

Common and different characteristics between PEA and Cholix are summarized in Table 1. Although PEA utilizes LRP1 as a receptor, Cholix binds to LRP1 and/or unknown receptor(s). Translocation of PEA and Cholix is commonly mediated by furin cleavage and KDEL pathways. Intraperitoneal administration of PEA and Cholix to mice results in liver damage. In human hepatocytes, Cholix, but not PEA, induces a decrease in PHB1 followed by PHB dysfunction.

The proposed cell death mechanism is shown in Figure 5. Cholix produced by NOVC binds to host cell receptor LRP1 and/or unknown cell surface receptor(s) and then its catalytic domain is translocated to the cytosol through furin cleavage and KDEL pathways. Cholix-induced cell death pathways varied among cell types. In HepG2 cells, TNF-α promoted Cholix-induced cell death, which includes PKC/MAPK activation, followed by induction of caspase-dependent apoptosis and a caspase-independent, yet to be defined, cell death signaling pathway. Human hepatocytes showed more sensitivity to Cholix than HepG2 cells. In Cholix-treated hepatocytes, mitochondrial dysfunction stimulates a caspase-dependent apoptotic signal pathway, which involves ROCK-1 cleavage and ROS generation. In HeLa cells, Cholix induced mitochondrial-dependent and -independent caspase activation. It remains unknown why ADP-ribosylation of eEF2 by Cholix resulted in activation of different signaling pathways among the various cell types. Further studies are required to define the mechanisms underlying Cholix cytotoxicity in different cell types.

## Figures and Tables

**Figure 1 toxins-13-00012-f001:**
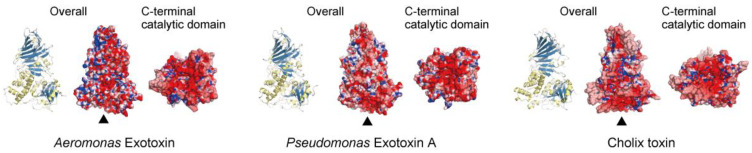
Crystal structures of three eEF2 mART cytotoxins. The overall structures (ribbon models and electrostatic surface models) and C-terminal catalytic domains (observed from the bottom, as indicated by triangles) of *Aeromonas* exotoxin (Protein Data Bank (PDB) ID 6Z5H), *Pseudomonas* exotoxin A (PDB ID 1IKQ), and Cholix toxin (PDB ID 3Q9O). The electrostatic potentials of molecular surfaces were calculated using Adaptive Poisson-Boltzmann Solver (APBS) [44]. The models were drawn by CueMol: Molecular Visualization Framework (http://www.cuemol.org/).

**Figure 2 toxins-13-00012-f002:**
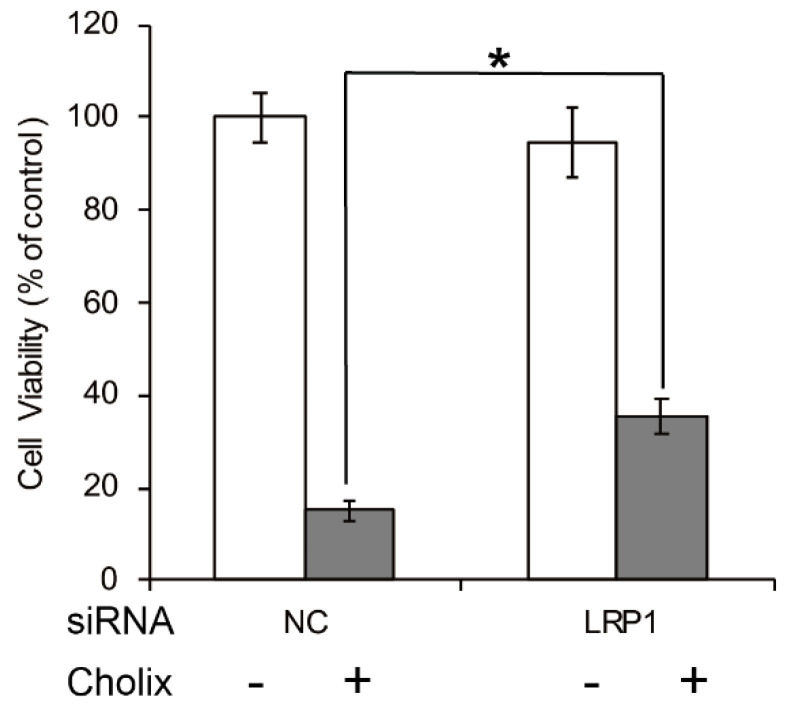
Effects of LRP1 knockdown in Cholix-treated HeLa cells. HeLa cells were treated with non-targeting control (NC) or LRP1 siRNA for 48 h and incubated with PBS (−) or Cholix (+) for 24 h. Cell viability was measured with a Cell Counting Kit (Dojindo). Asterisks indicate *p* < 0.05 with the Student’s *t*-test.

**Figure 3 toxins-13-00012-f003:**
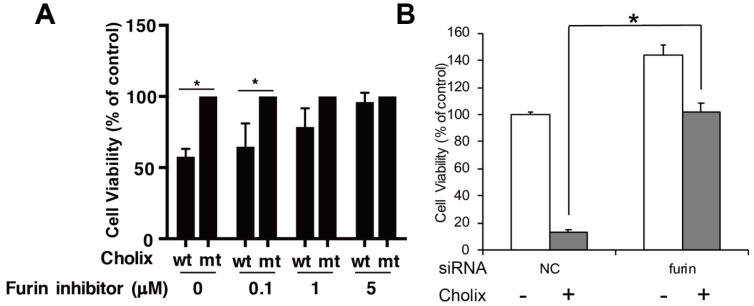
Effects of the furin inhibitor and its gene knockdown in Cholix-treated HeLa cells. (**A**) HeLa cells were incubated with catalytically inactive mutant Cholix (E581A) (mt) or wild-type Cholix (wt) in the presence of the indicated concentration of the furin inhibitor (decanoyl-RVKR-CMK). (**B**) HeLa cells were treated with non-targeting control (NC) or furin siRNA for 48 h and incubated with PBS (−) or Cholix (+) for 24 h. Cell viability was measured using a Cell Counting Kit. Asterisks indicate *p* < 0.05 with the Student’s *t*-test.

**Figure 4 toxins-13-00012-f004:**
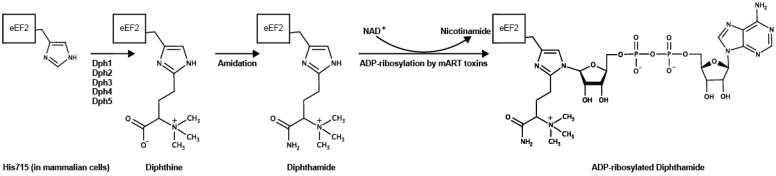
Diphthamide synthesis and ADP-ribosylation. His715 in mammalian (His699 in *Saccharomyces cerevisiae*) eEF2 is modified to diphthine by Dph family proteins (Dph1–5) followed by an amidation step using ATP and ammonia, presumably by Dph6 (in *Saccharomyces cerevisiae*) [69]. Diphtheria toxin, *Pseudomonas* exotoxin, *Aeromonas* exotoxin, and Cholix toxin ADP-ribosylate the diphthamide using NAD^+^ as an ADP-ribose donor. Marvin ChemAxon was used for drawing the formulas (https://www.chemaxon.com).

**Figure 5 toxins-13-00012-f005:**
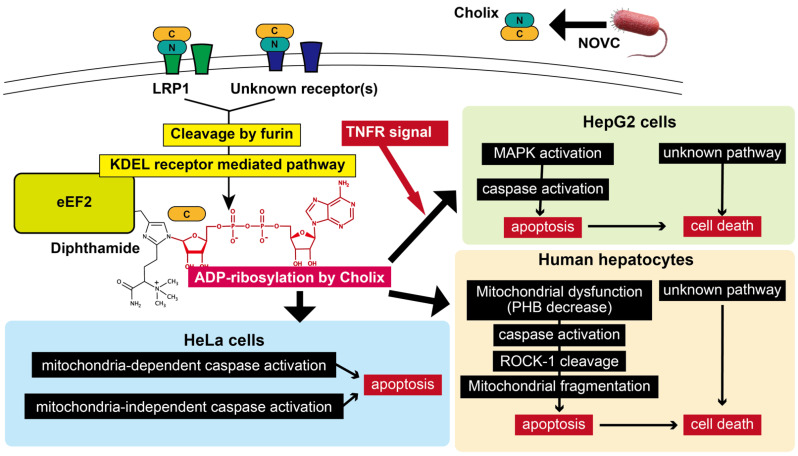
Proposed model of Cholix-induced cell death pathways. The cell death signal pathways in HepG2 cells [70], hepatocytes [46,70], and HeLa cells [47] are shown. This figure is described in the “Conclusion”. Marvin ChemAxon was used for drawing ADP-ribosylated diphthamide (https://www.chemaxon.com).

**Table 1 toxins-13-00012-t001:** Common and different characteristics between *Pseudomonas* exotoxin A and Cholix toxin [22,28,45,46,49,50,53].

Characteristics	*Pseudomonas* Exotoxin A	Cholix Toxin
Target	Diphthamide on eEF2	Diphthamide on eEF2
Enzymatic activity *K*_m(NAD+)_ (μm)/*k*_cat_ (s^−1^)	Mono-ADP-ribosylation 121 ± 21/13 ± 2	Mono-ADP-ribosylation 45 ± 3/10 ± 3
Host receptor	LRP1	LRP1 and/or unknown cell surface receptor(s)
Translocation	Furin protease-dependent KDEL motif (REDLK)-dependent	Furin protease-dependent KDEL motif (RKDELK)-dependent
Administration into mice	Liver damage	Liver damage
PHB1 expression	No change	Decrease

## Data Availability

Data are available upon request, please contact the contributing authors.

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
