# Peer review of "Cell Death Signaling Pathway Induced by Cholix Toxin, a Cytotoxin and eEF2 ADP-Ribosyltransferase Produced by Vibrio cholerae"

_toxins, 2020, doi:10.3390/toxins13010012_

Round 1

Reviewer 1 Report

Summary of findings

This paper reviews the literature surrounding the cell death signaling pathway induced by Cholix toxin. The authors also describe the cell death signaling pathways of Pseudomonas exotoxin A (PEA) and bits of information on other mono-ADP-ribosyltransferases (mART). Based on the reports referenced in the manuscript, the authors made many claims including: the probability of horizontal gene transfer is low from PEA, differences in NAD-binding pockets in the c-terminal domains could be responsible for differences in mART activity, there is a LRP1 independent Cholix receptor or intoxication pathway, Cholix follows a similar translocation pathway to that of PEA, and TNFa is an enhancesr of cholix-induced apoptosis.

General comment

The claims made in this review are novel and could provide viable knowledge to the field, however, these claims are too loosely made based on work that is largely observational and should be supported with scientific data.  Further, the manuscript dose not give enough information on the cellular mechanisms mentioned or in light of the claims made. The manuscript also jumps from cholix to PEA and other mARTs often making it hard to follow. While I do find the manuscript to have potential, in its current form, with the claims made, and lack of sufficient information/references it is not suitable for publication.

Author Response

Thank you very much for your thoughtful comments.

We have revised our manuscript.

Please see our attached file providing point-by-point responses.

Reviewer 2 Report

see attachment.

Author Response

(The authors gave the same response as above.)

Reviewer 3 Report

The review entitled “Cell death signaling pathway induced by Cholix toxin, a cytotoxin and eEF2 ADP-ribosyltransferase, produced by Vibrio cholerae” describes how the cholix cytotoxin interacts with target cells, related mechanisms underlying the action of the toxins and how the cellular pathways are modified by the toxin that acts as a mono-ADP-ribosyltransferase.

The manuscript is well organized and written in a clear manner; however, I found it a bit too concise, it is mostly a summary on “the state of the art” of the topic, with few interpretations. I would suggest to further comment the different points and, where possible, to be more detailed. For example, the authors did not comment a recent study describing the transcytosis of the toxin. (Taverner, Alistair, et al. "Cholix protein domain I functions as a carrier element for efficient apical to basal epithelial vvtranscytosis." Tissue Barriers 8.1 (2020): 1710429).

Moreover, I would also comment on the different pathways activated by the toxin according to the different cell types, and try to analyze (if possible) differences among toxin receptors on the different cell types.

In addition, I understand the scope of the study is to explain the different cellular pathways through which cell apoptosis occur, but mentioning the clinical relevance can increase the significance of the study.

Minor points: 1) NAD should be always indicated as NAD+

                      2) Figure 2, please correct “capase” with “caspase”.

Author Response

(The authors gave the same response as above.)

Round 2

Reviewer 2 Report

The revised manuscript is improved but still needs additional information, clarifications, and some minor corrections.

Additional information

Lines 68-69:  A reference after Fieldhouse et al is needed in the sentence "Based on Cholix/NAD+ complex structure and site-directed mutagenesis, Fieldhouse et al. proposed a new kinetic model of Cholix."  The authors should also explain what was proposed in the new kinetic model. 

Lines 69-70:  The authors should explain the mechanism(s) for compound-induced inhibition of Cholix.

Line 82:  Saelinger et al needs a reference.

Lines 110-111:  Was there any effect of LRP knockdown on Cholix activity in human hepatocytes?  The phrase "was not suppressed" is vague.  If there was a partial inhibitory effect on Cholix, the authors should state Cholix-induced apoptosis was only partially inhibited by LRP knockdown.  If there was no inhibitory effect, the authors should state Cholix-induced apoptosis was not suppressed at all.

Lines 119-120:  Does Cholix, like PEA, also have a disulfide bond that connects the furin-cleaved domains?

Clarifications

Lines 81-85:  The order of the final three sentences should be: 

Jørgensen et al. reported that the Km(NAD+) of Cholix is lower than that of PEA [22]. These differences might affect their mART activity in vivo.  While Saelinger et al. reported that the amount of toxin required to inhibit protein synthesis by 50 % (IC50) with PEA in mouse LM fibroblast was 32 ng/ml, Jørgensen et al. found that the IC50 of Cholix was 4.6 ± 0.4 ng/ml [22].

"These differences" (line 84) refers to the sentence ending "… lower than that of PEA [22]" and should therefore immediately follow that sentence.

Lines 116-120:  This section of the paragraph requires further clarification.  Cholix translocates from endosome to endoplasmic reticulum before reaching the cytosol.  The transition from the first sentence on Cholix to a discussion of PEA transport is abrupt.  The distinction between the roles of furin and protein disulfide isomerase need further clarification.  Suggestions:

"In host cells, Cholix is transported from endosome to endoplasmic reticulum and then reacts with eEF2 in the cytosol. PEA follows a similar intracellular path. In the acidic, early-endosomal environment, PEA is cleaved into two fragments by the protease furin, which which digests a furin-motif (RHRQPRG) [56,57] to generate a 28-kDa N-terminal domain and 37-kDa C-terminal mART domain.  Protein disulfide-isomerase further reduces a disulfide bond in PEA [58]."

Lines 143-144:  It is not clear how " Awasthi et al. proposed that Cholix may have functional capabilities to target diverse host systems" relates to the KDEL/HDEL discussion.  My original question about this topic was not addressed:  If the KDEL receptor recognizes both KDEL and HDEL sequences, why does a Cholix variant with HDELK in the place of the native KDELK sequence lack cytotoxicity? 

Lines 221-222:  "these different pathways" should be clarified.  Does this mean by the inhibition of protein synthesis combined with rapid proteasomal degradation?  I view these two coupled events as part of the same pathway. 

Minor corrections

Line 22:  mART needs to be defined in the abstract.

Line 87:  There are 4 eEF2 mART cytotoxins, so the "the" in the Figure title should be removed.

Line 161:  Diphtheria toxins should be Diphtheria toxin (no "s").

Line 199:  Cholix needs to be capitalized (sixth word on the line).

Line 219:  "which rapidly" should be "which is rapidly"

Line 246:  The additional, unknown receptor for Cholix could be a lipid.  There may also be more than one unknown receptor.  The phrase "unknown cell-surface protein" should therefore be replaced with "unknown cell-surface receptor(s)".  The "(s)" denotes there may be more than one unknown receptor and should also be used in Table 1 for the Cholix host receptor box.

Other information for the authors

In future work, the authors could use the furin-deficient LoVo cells to further demonstrate that furin is involved with Cholix activation.

The KDEL tag must be at the extreme C-terminus of a protein in order to be recognized by the KDEL receptor, so it is not surprising that a C-terminal GST tag would eliminate the cellular activity of Cholix.

Author Response

Thank you very much for your comments.

Please check my attached file.
